# Enhanced Photocatalytic Antibacterial Properties of TiO_2_ Nanospheres with Rutile/Anatase Heterophase Junctions and the Archival Paper Protection Application

**DOI:** 10.3390/nano11102585

**Published:** 2021-09-30

**Authors:** Yingying Qin, Xinyu Wang, Pengyuan Qiu, Jian Tian

**Affiliations:** 1Archives Department, China University of Petroleum (East China), Qingdao 266580, China; qinyingying@upc.edu.cn; 2School of Materials Science and Engineering, Shandong University of Science and Technology, Qingdao 266590, China; skd994440@sdust.edu.cn (X.W.); skd994462@sdust.edu.cn (P.Q.)

**Keywords:** TiO_2_ nanospheres, rutile, anatase, photocatalytic antibacterial activity, archival paper protection

## Abstract

TiO_2_ has been generally studied for photocatalytic sterilization, but its antibacterial activities are limited. Herein, TiO_2_ nanospheres with rutile/anatase heterophase junctions are prepared by a wet chemical/annealing method. The large BET surface area and pore size are beneficial for the absorption of bacteria. The rutile/anatase heterojunctions narrow the bandgap, which enhances light absorption. The rutile/anatase heterojunctions also efficiently promote the photogenerated carriers’ separation, finally producing a high yield of radical oxygen species, such as •O_2_^−^ and •OH, to sterilize bacteria. As a consequence, the obtained TiO_2_ nanospheres with rutile/anatase heterojunctions present an improved antibacterial performance against *E. coli* (98%) within 3 h of simulated solar light irradiation, exceeding that of TiO_2_ nanospheres without annealing (amorphous) and TiO_2_ nanospheres annealing at 350 and 550 °C (pure anatase). Furthermore, we design a photocatalytic antibacterial spray to protect the file paper. Our study reveals that the TiO_2_ nanospheres with rutile/anatase heterojunctions are a potential candidate for maintaining the durability of paper in the process of archival protection.

## 1. Introduction

Archives are a non-renewable important source. Archival entities, like other materials, will be inevitably degraded and aged with the prolongation of saving time. Moreover, microorganisms and bacteria in the air of the archive store can fall into the archival entity, thereby affecting the durability of the archival entity. Paper is one of the most important carrier materials in existing archival entities. In the process of archival protection, maximizing the maintenance of paper’s durability is vital.

Traditional disinfection methods, such as UV disinfection, ozonation, and chlorination, are widely used [1]. However, they have the disadvantage of being operationally intensive, having an energy cost, and forming hazardous disinfection byproducts [2]. However, photocatalytic antibacterial technology using sustainable solar energy with no disinfection byproducts is an attractive technology for oxidative disinfection to various bacteria [3]. TiO_2_ as a photocatalyst is widely studied owing to being chemically stable and inexpensive [4]. After light irradiation, the nearby O_2_ molecules can react with the electrons to form •O_2_^−^. Meanwhile, the H_2_O molecules can capture holes to form •OH. •O_2_^−^ and •OH are highly reactive oxygen species (ROS); these ROS have strong oxidation potentials and can degrade bacteria [5]. Yet, the photocatalytic antibacterial performance of TiO_2_ is limited due to the fast carrier recombination [6]. To improve TiO_2_’s photocatalytic performance, reducing the carrier recombination is needed.

Semiconductor heterojunctions, such as band-structure matching heterojunction, p–n junction, and Schottky junction, are designed to enhance the carrier separation [7]. TiO_2_ possesses rutile, brookite, TiO_2_(B), and anatase polymorphs [8]. TiO_2_(B) and anatase in TiO_2_ nanobelts present different band structures and form a heterophase junction, which can promote the separation of photoinduced carriers [9]. Among the four polymorphs, rutile and anatase are the most studied in photocatalysis [10], which can also construct the heterophase junction and promote carrier separation [11]. The preparation of TiO_2_ nanomaterials can generally be divided into four strategies, including the hydrothermal strategy, sol–gel strategy, template strategy, and template-free (wet chemical) strategy [12]. The hydrothermal strategy needs high production costs and strict equipment requirements for high pressure and high temperature [13]. The sol–gel strategy easily causes the agglomeration of TiO_2_ [14]. However, the template-free (wet chemical) strategy only requires one step of reaction, such as annealing, which avoids the impurities and the affecting of the inherent properties and structural stability of the material caused by template methods [15]. Thus, the template-free (wet chemical) strategy is more effective and flexible for large-scale manufacturing than other methods [13].

In this paper, we use a wet chemical/annealing method to prepare TiO_2_ nanospheres with rutile/anatase heterophase junctions. The construction of rutile/anatase heterophase junctions in TiO_2_ nanospheres and large BET surface areas/pore sizes can efficiently improve the light absorption, enhance the absorption of bacteria, and promote carrier separation. Hence, compared with amorphous TiO_2_ (without annealing) and pure anatase TiO_2_ (annealing at 350 and 550 °C), the obtained TiO_2_ nanospheres with rutile/anatase heterojunctions (annealing at 750 °C) present the enhanced photocatalytic antibacterial performance against *Escherichia coli* (*E. coli*). Inspired by this, we design a photocatalytic antibacterial spray with TiO_2_ nanospheres with rutile/anatase heterojunctions as the photocatalysts. The TiO_2_ nanospheres with rutile/anatase heterojunctions spray can maintain the durability of paper in the process of archival protection.

## 2. Experimental Section

### 2.1. Materials

Tetrabutoxytitanium (Ti(OC_4_H_9_)_4_, TBOT, 99%), ammonium hydroxide (NH_3_·H_2_O, 25 wt%), and epichlorohydrin (98%) were provided by Sinopharm (Beijing, China). The paper used in the experiment was Double-A A4 white copy paper (70 g/m^2^, Yiwang Paper Co., LTD, Shanghai, China).

### 2.2. Synthesis of TiO_2_ Nanospheres

Amorphous TiO_2_ nanospheres were firstly synthesized by a wet chemical procedure. Twenty milliliters of Ti(OC_4_H_9_)_4_ (TBOT) as the Ti source was quickly injected into 200 mL deionized water with 6 g NH_3_·H_2_O. After reacting for 30 s, the white precipitate was washed thoroughly and dried at 60 °C for 10 h. The amorphous TiO_2_ (TiO_2_-A) was obtained. Thermal annealing of the amorphous TiO_2_ at 250, 350, 550, and 750 °C for 4 h led to the production of TiO_2_ nanospheres, labeled as TiO_2_-250, TiO_2_-350, TiO_2_-550, and TiO_2_-750, respectively.

### 2.3. Characterization

XRD patterns were carried out using a D8 Advance (Bruker, Berlin, Germany) powder X-ray diffractometer. TEM and SEM with an EDS were performed using a JEM 2100F (JOEL, Tokyo, Japan) and NanoSEM 450 (FEI, Portland, OR, USA) microscope, respectively. XPS was performed with the ESCALAB 250 instrument. The UV-Vis DRS spectra were examined on the UV-3101 UV-Vis spectrophotometer (SHIMADZU, Tokyo, Japan). The BET specific surface area was measured with the ASAP2020 (Micromeritics, Norcross, GA, USA) instrument. The electric paper board breaking resistance tester (Yq-zb-1) was manufactured by Hangzhou Light Industry Testing Instrument Co., LTD, Hangzhou, China.

### 2.4. Photocatalytic Activity Tests

*Escherichia coli* (*E. coli*, ATCC 25922) was selected for photocatalytic sterilization, in which the Xe arc lamp (300 W) with a simulated solar light filter was the simulated solar light source. The bacterial suspension was diluted to ~5 × 10^4^ colony forming units (CFU)/mL, and 100 μL of which was mixed with TiO_2_ nanospheres (each at 30 mg) in a test tube containing 10 mL of distilled water. Then, a 0.1 mL aliquot of the bacterial suspension mixtures was collected at given time intervals after the light turned on, diluted with the PBS buffer solution, and then spread on nutrient agar at 37 °C for 24 h. Error bars represent the standard deviation of three independent experiments.

### 2.5. Paper Breakage Resistance Tests

Potassium permanganate and formaldehyde fumigation: We adopted a 40% formaldehyde solution of 3 mL/m^3^ and potassium permanganate of 1.5 g/m^3^ to calculate the amount of disinfection in the archives, and the fumigation time was 15 h. The doors and windows were kept closed during disinfection. The doors and windows were opened for 48 h after the fumigation.

Photocatalytic bactericidal spray: The photocatalytic bactericidal spray was composed of photocatalysts (TiO_2_-750) and epichlorohydrin, in which water:TiO_2_-750:epichlorohydrin = 1000:1:5. The photocatalytic bactericidal spray solution was also 3 mL/m^3^ to calculate the amount of disinfection in the archives, and disinfection time was 15 h. The doors and windows were kept closed during disinfection. The doors and windows were opened for 48 h after the disinfection.

The same tested paper was added 3 days before disinfection. After disinfection for 30 days, the paper was sealed in a sealed bag and sent to the laboratory for testing. The reference standard for breakage resistance detection was the Determination of paper breakage resistance (GB/T 454-2002).

## 3. Results and Discussion

As shown in Figure 1, without annealing (TiO_2_-A, curve black) and after annealing at 250 °C (TiO_2_-250, curve red), no obvious diffraction peaks are observed, which indicates that TiO_2_-A and TiO_2_-250 are amorphous structures. After annealing at 350 °C (TiO_2_-350, curve blue) and 550 °C (TiO_2_-550, curve green), the diffraction peaks at about 2θ = 25.28°, 37.79°, 48.05°, 53.88°, 55.06°, and 62.69° are ascribed to (101), (004), (200), (105), (211), and (204) planes of anatase TiO_2_ (JCPDS card no. 21-1272) [16], suggesting that TiO_2_-350 and TiO_2_-550 exist in a pure anatase phase. After annealing at 750 °C (TiO_2_-750, curve purple), besides anatase diffraction peaks, the diffraction peaks at about 2θ = 27.49°, 36.15°, and 54.43° are assigned to the rutile TiO_2_ (JCPDS card no. 21-1276) [17], indicating that TiO_2_-750 is mixed-phase (anatase and rutile) crystalline. According to the peak areas of TiO_2_-750, the proportion of anatase (69.5%) and rutile (30.5%) phase is obtained. As the calcination temperature increases from 350 to 750 °C, the diffraction peaks’ intensities become stronger due to the increase of crystallinity upon increasing calcination temperature.

For the XPS survey spectrum (Appendix A), besides O 1s and Ti 2p peaks, the C 1s peak at 288.3 eV is detected, due to the adventitious hydrocarbon from the air [18]. For the Ti 2p spectrum (Figure 2a), the two main peaks of Ti2p_1/2_ at 464.4 eV and Ti2p_3/2_ at 458.6 eV are ascribed to Ti^4+^ of TiO_2_ [19]. Furthermore, an associated satellite peak located at 458.2 eV is detected [20], corresponding to Ti^3+^. The O 1s spectrum of TiO_2_ (Figure 2b) can be fitted with three peaks at 529.9, 531.2, and 532.0 eV, ascribing to O^2−^ (O-Ti-O bonds), O^−^ (Ti^3+^ induced oxygen vacancies), and Ti-OH groups, respectively [21]. Furthermore, the ratio of Ti^4+^ and Ti^3+^ in the TiO_2_-750 sample is 91.5:8.5, based on their characteristic peaks in XPS spectra. Based on the above-discussed analysis, the XPS results confirm the existence of oxygen vacancies in TiO_2_-750. Calcination is the most common approach for forming TiO_2_ defects, such as oxygen vacancies [21]. Thermal annealing of the amorphous TiO_2_ at 750 °C for 4 h led to the production of oxygen vacancies in TiO_2_-750.

The microstructure of TiO_2_ synthesized with and without annealing was analyzed by SEM and TEM (Figure 3 and Figure 4). Figure 3 presents the SEM images of TiO_2_. TiO_2_ without annealing (TiO_2_-A, Figure 3a) presents the sphere structure with 200–300 nm in diameter. With the increase of annealing temperature from 250 to 700 °C (Figure 3b–e), the change of morphology of TiO_2_ nanospheres is not obvious.

The TEM image of TiO_2_-750 presents that TiO_2_ nanospheres are composed of nanosheets with the diameter of 50–100 nm (Figure 4a) and the thickness of 10–20 nm (Figure 4b). The HRTEM image in Figure 4b presents that TiO_2_ nanospheres are extremely crystallized. The crystal lattice of 0.357 and 0.320 nm indicate anatase TiO_2_ (101) facets and rutile TiO_2_ (110) facets, respectively [10]. The anatase and rutile phases in TiO_2_ nanospheres can form the heterophase junctions, which will efficiently enhance the separation of photogenerated carriers. The EDS elemental mapping images (Appendix A) reveal the existence of Ti and O elements, further confirming the formation of TiO_2_.

The specific surface areas were tested by nitrogen adsorption–desorption isotherms. The BET surface area of TiO_2_ nanospheres is significantly increased after annealing (Appendix A). The annealing temperature does not affect the specific surface area of TiO_2_ nanospheres. TiO_2_-750 still presents a relatively large specific surface area (297 m^2^/g). Interestingly, TiO_2_-750 (Appendix A) presents the largest pore diameter (4.2 nm), which is beneficial for the absorption of bacteria.

As shown in Figure 5a, the TiO_2_-A only shows the absorption of UV light and exhibits absorption edges at about 375 nm. Interestingly, after annealing at 750 °C, the absorption edge of TiO_2_-750 is expanded to 442 nm, indicating that the formation of the rutile/anatase junction and oxygen vacancies in TiO_2_-750 result in a longer-wavelength (visible) absorption edge and have significant effects on the optical property. Figure 5b shows the calculated bandgap (*E*_g_) for TiO_2_ nanospheres. The calculated *E*_g_ of TiO_2_-A, TiO_2_-250, TiO_2_-350, TiO_2_-550, and TiO_2_-750 is estimated to be 3.30, 3.10, 3.05, 3.00, and 2.80 eV, respectively. Interestingly, TiO_2_-750 exhibits the narrowest bandgap, implying the best light absorption.

To prove the photocatalytic antibacterial performance of TiO_2_ nanospheres, the sterilization of *E. coli* was measured under simulated solar light irradiation (Figure 6a). In the light control experiment (Appendix A), the cell density of *E. coli* shows a limited decrease without photocatalysts under simulated solar light irradiation, suggesting the negligible effect of the simulated solar light on the bacterial cells. The TiO_2_ nanosphere without annealing (TiO_2_-A) is not a good photocatalyst (Figure 6a). The sterilization rate of TiO_2_-A for 120 min is only 12%. After annealing, the photocatalytic antibacterial activity of TiO_2_ nanospheres is significantly improved. Compared with other annealing temperature samples (TiO_2_-250, TiO_2_-350, and TiO_2_-550), TiO_2_ nanosphere annealing at 750 °C (TiO_2_-750) presents the best photocatalytic antibacterial activity. After irradiation for 120 min, the *E. coli* sterilization rate of TiO_2_-750 is 98%. This is ascribed to the formed rutile/anatase heterojunctions, which effectively promote the separation of carriers. In addition, the relatively larger specific surface area (297 m^2^ g^−1^) and the largest pore diameter (4.2 nm) of TiO_2_-750 are beneficial for the absorption of bacteria. Moreover, TiO_2_-750 has the narrowest bandgap and can efficiently utilize more light. Moreover, there is no significant loss of *E. coli* sterilization rate over TiO_2_-750 after three cycles (Appendix A), suggesting excellent sterilization stability.

To confirm the photocatalytic antibacterial mechanism of TiO_2_-750, three main reactive species (•O_2_^−^, •OH, and h^+^) in the photocatalytic sterilization were measured using the radical scavenger experiments (Figure 6b). Benzoquinone (1 mM), methanol (1:15/V:V), and EDTA (10 mM) were used as •O_2_^−^, •OH, and h^+^ radical scavengers, respectively. After adding EDTA, the photocatalytic efficiency is slightly decreased compared with no trapping agent, indicating that h^+^ has a slight effect on photocatalytic sterilization. The addition of methanol and benzoquinone significantly reduces photocatalytic sterilization, indicating that •O_2_^−^ and •OH are the main active substances for the sterilization of *E. coli* in TiO_2_-750.

The photocatalytic antibacterial mechanism is proposed in Figure 7. The rutile phase has upper positions of both VB and CB than anatase [22]. For TiO_2_-750 with rutile/anatase heterojunctions (Figure 7a), during the photocatalytic sterilization, the electrons and holes of both anatase and rutile are excited (Equation (1)). The photoinduced electrons in the CB of rutile are migrated to the anatase and react with dissolved O_2_ to produce •O_2_^−^ (Equation (2)). In the meantime, the holes in the VB of anatase are transferred to rutile and captured by H_2_O to produce •OH (Equation (3)), finally inactivating *E. coli* cells.
Catalysts + hν → h^+^ + e^−^(1)
e^−^ + O_2_ → •O_2_^−^(2)
h^+^ + H_2_O → •OH(3)

The strong oxidizing reactive oxygen species (ROS), such as •O_2_^−^ and •OH, are generated in the TiO_2_-750 with rutile/anatase heterojunctions, confirmed by the reactive species trapping experiments (Figure 6b), triggering oxidative stress and cell damage in bacteria [23].

However, for the pure anatase phase (TiO_2_-350 and TiO_2_-550) (Figure 7b), under light irradiation, the holes and electrons are generated (Equation (1)). The holes and electrons can be quickly recombined, which induces the low photocatalytic performance.

Inspired by the excellent photocatalytic antibacterial activity of TiO_2_-750 with rutile/anatase heterojunctions, we designed a photocatalytic bactericidal spray (water:TiO_2_-750:epichlorohydrin = 1000:1:5) with TiO_2_-750 with rutile/anatase heterojunctions as the photocatalysts to protect the file paper. As shown in Figure 8, the photocatalytic bactericidal spray method has a positive effect on the paper breakage resistance, and the breakage resistance decreases 7.5% after photocatalytic bactericidal spraying after 30 days, which is better than the paper without photocatalytic bactericidal spraying (decreasing 24.6% after 30 days) and the paper fumigated with potassium permanganate and formaldehyde (decreasing 18.7% after 30 days). Our designed photocatalytic bactericidal spray using TiO_2_-750 with rutile/anatase heterojunctions as the photocatalysts can maintain the durability of paper in the process of archival protection.

## 4. Conclusions

In conclusion, TiO_2_ nanospheres with rutile/anatase heterojunctions are successfully fabricated. With the merits of improved carrier transfer by rutile/anatase heterojunctions, and good adsorption performance for bacteria by large specific surface area and pore size, the photocatalytic antibacterial efficiency of TiO_2_ nanospheres with rutile/anatase heterojunctions towards *E. coli* is significantly improved. Under light illumination, the photoinduced electrons in the CB of rutile are immigrated to the anatase through rutile/anatase heterojunctions. Meanwhile, the holes in the anatase’s VB are transferred to rutile. These photo-excited electrons and holes can react with O_2_ and H_2_O to produce a great quantity of ROS (e.g., •O_2_^−^ and •OH), which can induce serious damage to bacterial membranes, thus resulting in bacterial death. Inspired by the excellent photocatalytic antibacterial activity of TiO_2_ nanospheres with rutile/anatase heterojunctions as the photocatalysts to protect the file paper, the paper breakage resistance is significantly improved after spraying with the photocatalysts. Therefore, it is a potential candidate for maintaining the durability of paper in the process of archival protection.

## Figures and Tables

**Figure 1 nanomaterials-11-02585-f001:**
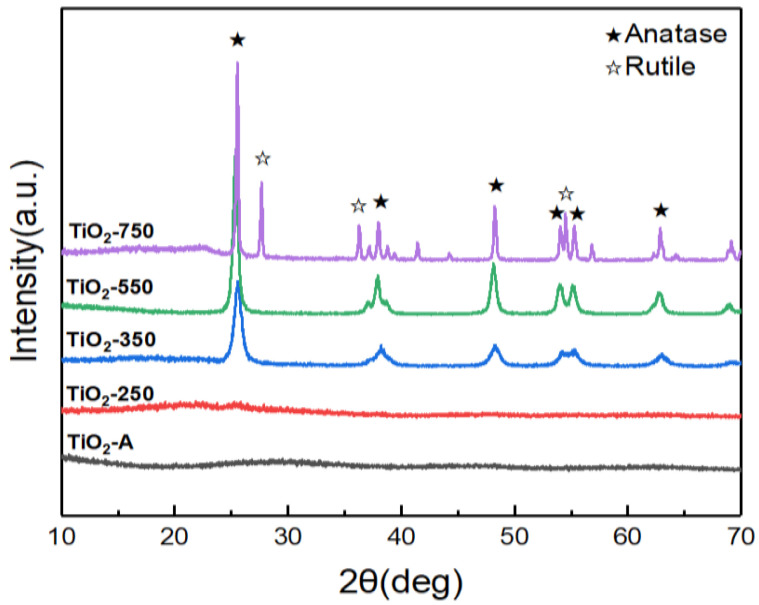
XRD patterns of TiO_2_-A, TiO_2_-250, TiO_2_-350, TiO_2_-550, and TiO_2_-750.

**Figure 2 nanomaterials-11-02585-f002:**
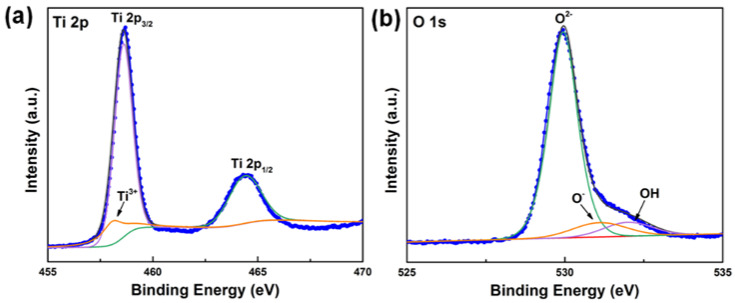
(**a**) Ti 2p and (**b**) O 1s XPS spectra of TiO_2_-750.

**Figure 3 nanomaterials-11-02585-f003:**
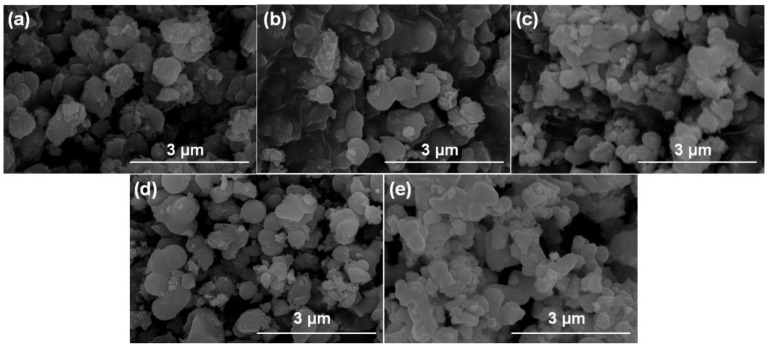
SEM images of (**a**) TiO_2_-A, (**b**) TiO_2_-250, (**c**) TiO_2_-350, (**d**) TiO_2_-550, and (**e**) TiO_2_-750.

**Figure 4 nanomaterials-11-02585-f004:**
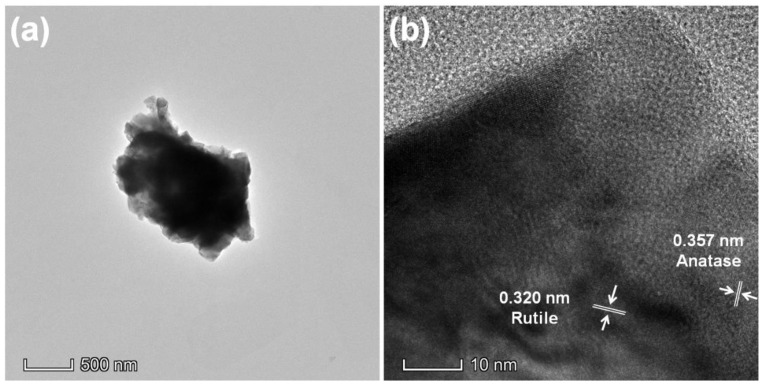
(**a**) TEM and (**b**) HRTEM images of TiO_2_-750.

**Figure 5 nanomaterials-11-02585-f005:**
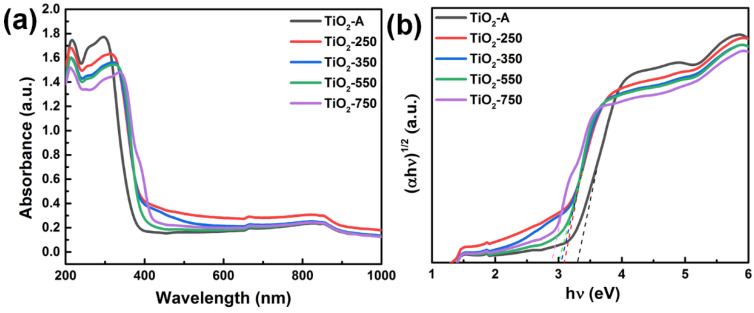
(**a**) UV-Vis absorption spectra and (**b**) bandgap value of TiO_2_-A, TiO_2_-250, TiO_2_-350, TiO_2_-550, and TiO_2_-750.

**Figure 6 nanomaterials-11-02585-f006:**
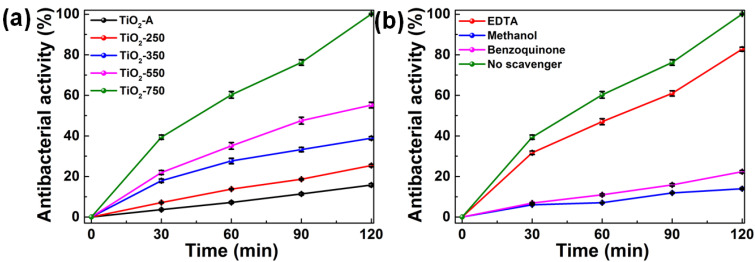
(**a**) Photocatalytic antibacterial efficiency against *E. coli* for TiO_2_-A, TiO_2_-250, TiO_2_-350, TiO_2_-550, and TiO_2_-750 under simulated solar light irradiation; (**b**) reactive species trapping experiments of TiO_2_-750 under simulated solar light irradiation. Error bars represent the standard deviation of three independent experiments.

**Figure 7 nanomaterials-11-02585-f007:**
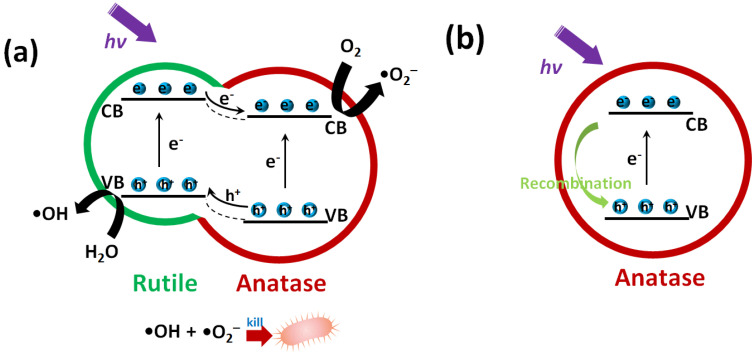
The photocatalytic antibacterial mechanism of TiO_2_ nanospheres (**a**) with and (**b**) without rutile/anatase heterojunctions. Adapted with permission from ref. [10]. 2020 Elsevier.

**Figure 8 nanomaterials-11-02585-f008:**
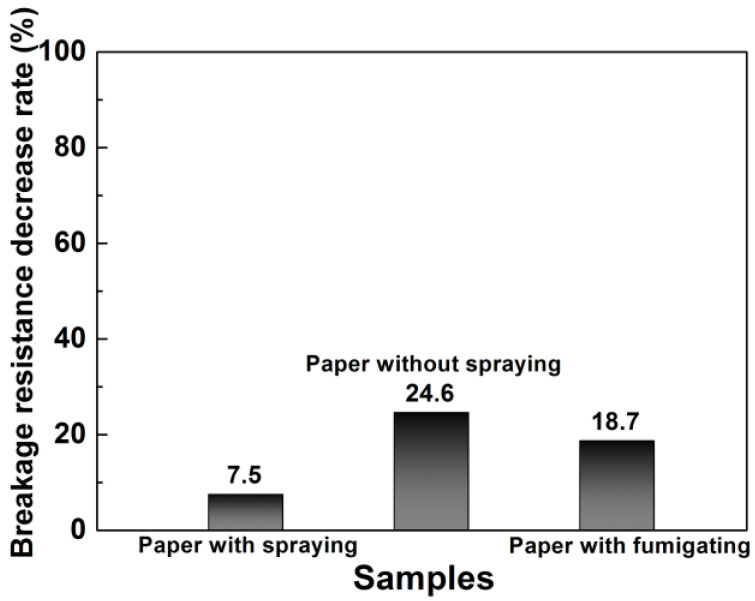
The impact of paper breaking resistance with and without the photocatalytic bactericidal spray, with potassium permanganate and formaldehyde fumigation.

## Data Availability

Not applicable.

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
