# Peer review of "Enhanced Photocatalytic Antibacterial Properties of TiO2 Nanospheres with Rutile/Anatase Heterophase Junctions and the Archival Paper Protection Application"

_nanomaterials, 2021, doi:10.3390/nano11102585_

Round 1

Reviewer 1 Report

I consider that the manuscript should be reviewed before sending for publication. I send my comments in a pdf file.

Author Response

Reviewer #1:

Response: Thank the reviewer very much for your instructions on the revision of our manuscript. We greatly appreciate the insightful comments of the reviewer of the paper. You have made some valuable suggestions that have led to great improvements to the manuscript.

1. There are many English writing errors. The document presents some flaws and should not be sent to a revision process before it has been carefully written and formatted. I will try to point out some of the flaws that I found while reading the manuscript sent.

Response: Thanks for the editor’s detailed review. The mistakes including spelling and grammar have been carefully checked and corrected.

2. The idea of using photocatalytic experiments on archival paper is controversial since UV irradiation has other side effects that prevent its usage in such applications. Nevertheless, the idea behind the manuscript may be useful. The evidence of template-free method of synthesis seems inappropriate. It is reported a simple wet chemical synthesis, but it is not clear the idea of a template free method and why this terminology was chosen.

Response: Thanks for the editor’s suggestion. The description of template-free method of synthesis is not correct, we have revised it into wet chemical method.

3. From XRD the authors conclude the existence of rutile phase upon annealing at 750 ºC, stating it as “As the annealing temperature increase from 350 ºC to 750 ºC, the diffraction peaks’ intensities become stronger”. This sentence is obvious but further analysis could had been performed and the authors do not explain what can happen to the structure or at what temperature range the phase transformation should occur. It shows a clear evidence of the increase of crystallinity upon increasing annealing temperature, as expected, but it is not properly mentioned or discussed. This analysis should be more carefully presented.

Response: Thanks for the editor’s detailed review. As shown in Figure 1 in the paper, without annealing (TiO2-A, curve black) and after annealing at 250 oC (TiO2-250, curve red), no obvious diffraction peaks are observed, indicate that TiO2-A and TiO2-250 are amorphous structures. After annealing at 350 oC (TiO2-350, curve blue) and 550 oC (TiO2-550, curve green), the diffraction peaks are ascribed to anatase TiO2 (JCPDS card no. 21-1272), suggest that TiO2-350 and TiO2-550 exist in pure anatase phase. After annealing at 750 oC (TiO2-750, curve purple), besides anatase diffraction peaks, the diffraction peaks at about 2θ=27.49°, 36.15° and 54.43° are assigned to the rutile TiO2 (JCPDS card no. 21-1276), indicate that TiO2-750 is mixed-phase (anatase and rutile) crystalline. According to the peak areas of TiO2-750, the proportion of anatase (69.5%) and rutile (30.5%) phases is obtained. As the calcination temperature increase from 350 oC to 750 oC, the diffraction peaks’ intensities become stronger due to the increase of crystallinity upon increasing calcination temperature.

4. From XPS apart than the presence of oxygen vacancies in the sample annealed at 750ºC, nothing else is concluded. There is no clear reason why these results are presented and no comparison with samples annealed at different annealing temperature is presented. As such this section seems irrelevant to understand if the sample annealed at 750 ºC has Oxygen vacancies and to understand the final behaviour.

Response: Thanks for the reviewer’s good question. Calcination is the most common approach for engineering TiO2 defects, such as oxygen vacancies [ ACS Nano 2017, 11, 821–830]. The oxygen vacancies are caused by an oxygen escape. As shown in Figure 2b, The O 1s spectrum of TiO2 can be fitted with three peaks at 529.9, 531.2 and 532.0 eV, ascribing to O2- (O-Ti-O bonds), O- (Ti3+ induced oxygen vacancies) and Ti-OH groups, respectively. Thermal annealing of the amorphous TiO2 at 750 oC for 4 h led to the production of oxygen vacancies in TiO2-750.

5. The sentence “ With the increase of annealing temperature from 250 oC to 700 oC (Figure 3b-e), the change of structure and morphology of TiO2 nanospheres is not obvious.” Should be re-written. If figure 3, the authors can only rely on morphological properties, since through XRD the structure and increase of intensity and decrease of width of the diffraction peaks (not discussed by the authors) already indicated structural modification of the samples.

Response: Thanks for the editor’s detailed review. The sentence “With the increase of annealing temperature from 250 oC to 700 oC (Figure 3b-e), the change of structure and morphology of TiO2 nanospheres is not obvious.” has been revised into “With the increase of annealing temperature from 250 oC to 700 oC (Figure 3b-e), the change of morphology of TiO2 nanospheres is not obvious.”

6. Despite the authors mention the material as nanospheres when referring to TEM, the notion of nanosheets is also present, however no specific dimensions as thicknesses are referred. If nanosheets are to be considered and indicated, the authors are encouraged to refer the thickness of these sheets. From SEM analysis however other shapes are also observed so the authors are encouraged to be coherent Furthermore, it is hard from the image shown on Figure 4 to conclude this morphology.

Response: Thanks for the reviewer’s suggestion. The HRTEM image in Figure 4b presents that the thickness of nanosheet is about 10-20 nm. We have re-measured the TEM (Figure 4a) and obtained that TiO2 nanospheres are composed of nanosheets.

7. The antibacterial experiments were not clear. It cannot be understood if the authors used the nanospheres as powder to see the photocatalytic effect, and if so, how were they used, in what conditions and quantities. The experimental section is really poor, and it should carefully describe the experiment steps.

Response: Thanks for the reviewer’s suggestion. The antibacterial experiments have been revised in the experimental section.

Escherichia coli (E. coli, ATCC 25922) was selected for photocatalytic sterilization, in which the Xe arc lamp (300 W) with a simulated solar light filter as the simulated solar light source. The bacterial suspension was diluted to ~5 × 104 colony forming units (CFU)/mL, and 100 μL of which was mixed with TiO2 nanospheres (each at 30 mg) in a test tube containing 10 mL of distilled water. 0.1 mL aliquot of the bacterial suspension mixtures were collected at given time intervals after the light turned on, diluted with the PBS buffer solution, and then spread on nutrient agar at 37 oC for 24 h.

8. The mechanisms are well presented although not very Novell.

Response: Thanks for the reviewer’s suggestion. The photocatalytic mechanism has been revised according to the reviewer’s suggestion.

The photocatalytic antibacterial mechanism is proposed (Figure 7). The rutile phase has upper positions of both VB and CB than anatase [18]. For TiO2-750 with rutile/anatase heterojunctions (Figure 7a), during the photocatalytic sterilization, the electrons and holes of both anatase and rutile are excited (Eq. (1)). The photoinduced electrons in the CB of rutile are migrated to the anatase and react with dissolved O2 to produce •O2(Eq. (2)). In the meantime, the holes in the VB of anatase are transferred to rutile and captured by H2O to produce •OH (Eq. (3)), finally inactivating E. coli cells.

Catalysts + hν → h+ + e-                           (1)

e- + O2 → •O2                            (2)

h+ + H2O → •OH                            (3)

The strong oxidizing reactive oxygen species (ROS), such as •O2 and •OH, are generated in the TiO2-750 with rutile/anatase heterojunctions, confirmed by the reactive species trapping experiments (Figure 6b), triggering oxidative stress and cell damage in bacteria [19].

However, for the pure anatase phase (TiO2-350 and TiO2-550) (Figure 7b), under light irradiation, the holes and electrons are generated (Eq. (1)). The holes and electrons can be fast recombined, which induced the low photocatalytic performance.

9. The design of the photocatalytic bactericidal spray should also appear on the experimental section. It is really confuse and makes it hard to understand the procedure followed by the authors.

Response: Thanks for the reviewer’s suggestion. The design of the photocatalytic bactericidal spray has been added in the experimental section.

The paper used in the experiment is Double-A A4 white copy paper (70g/m2, Yiwang Paper Co., LTD). Electric paper board breaking resistance tester (Yq-zb-1) is manufactured by Hangzhou Light Industry Testing Instrument Co., LTD.

2.5 Paper breakage resistance test

Potassium permanganate and formaldehyde fumigation: We adopt 40% formaldehyde solution of 3 mL/m3 and potassium permanganate 1.5 g/m3 to calculate the amount of disinfection in the archives, fumigation time is 15h. Keep doors and windows closed during disinfection. Open the door and window for 48h after the fumigation.

Photocatalytic bactericidal spray: The photocatalytic bactericidal spray is composed of photocatalysts (TiO2-750) and epichlorohydrin, in which water : TiO2-750 : epichlorohydrin =1000:1:5. The photocatalytic bactericidal spray solution is also 3 mL/m3 to calculate the amount of disinfection in the archives, disinfection time is 15h. Keep doors and windows closed during disinfection. Open the door and window for 48h after the disinfection.

The same tested paper will be added 3 days before disinfection. After disinfection for 30 days, the paper is sealed in a sealed bag and sent to the laboratory for testing. The reference standard for breakage resistance detection is the Determination of paper breakage resistance (GB/T 454-2002).

10. There are also some errors in the conclusions, such as: which can induce serious damage of bacterial membranes, resulting in bacterial death.

Response: Thanks for the editor’s detailed review. The mistakes including spelling and grammar have been carefully checked and corrected.

Reviewer 2 Report

In this paper, the authors prepared the TiO2 nanospheres with anatase/rutile heterophase junctions via a template-free/annealing method. Also, the authors design a photocatalytic antibacterial spraying to protect the file paper. The results of this paper are good presented. For this paper, the answer is “Minor Revision”. The authors must include:

  • At Introduction, please include more information’s about the advantages and disadvantage of template-free/annealing method in comparison with other methods reported in literature (e.g., in-situ phase transformation, hydrothermal/calcination method)
  • At Section 2.1, for all reagents indicates the manufacturer and concentration (if it is possible). Also, indicate the role of reagents used in synthesis.

Author Response

Reviewer #2:

Response: Thank the reviewer very much for your instructions on the revision of our manuscript. We greatly appreciate the insightful comments of the reviewer of the paper. You have made some valuable suggestions that have led to great improvements to the manuscript.

1. At Introduction, please include more information’s about the advantages and disadvantage of template-free/annealing method in comparison with other methods reported in literature (e.g., in-situ phase transformation, hydrothermal/calcination method).

Response: Thanks for the reviewer’s suggestion. The advantages and disadvantage of template-free/annealing method in comparison with other methods reported in literature have been added in the introduction of the revised manuscript.

The preparation of TiO2 nanomaterials can generally be divided into four strategies, including hydrothermal strategy, sol-gel strategy, template strategy, and template-free (wet chemical) strategy. The hydrothermal strategy needs high production costs and strict equipment requirements for high pressure and high temperature. The sol-gel strategy is easy to cause the agglomeration of TiO2. However, the template-free (wet chemical) strategy only requires one step of reaction such as annealing, which avoiding the impurities and affecting of the inherent properties and structural stability of the material caused by template methods. Thus, the template-free (wet chemical) strategy is more effective and flexible for large-scale manufacturing than other methods.

2. At Section 2.1, for all reagents indicates the manufacturer and concentration (if it is possible). Also, indicate the role of reagents used in synthesis.

Response: Thanks for the reviewer’s suggestion. the manufacturer and concentration of all reagents have been added in the Experimental Section of the revised manuscript. The role of reagents used in synthesis are also added (please see the Experimental Section of the revised manuscript).

2.1 Materials

Tetrabutoxytitanium (Ti(OC4H9)4, TBOT, 99%), ammonium hydroxide (NH3·H2O, 25 wt%) and epichlorohydrin (98%) were provided by Sinopharm. The paper used in the experiment was Double-A A4 white copy paper (70g/m2, Yiwang Paper Co., LTD).

2.2 Synthesis of TiO2 nanospheres

Amorphous TiO2 nanospheres were firstly synthesized by a wet chemical procedure. 20 mL Ti(OC4H9)4 (TBOT) as the Ti source was quickly injected 200 mL deionized water with 6 g NH3·H2O. After reacting for 30 s, the white precipitate was washed thoroughly and dried at 60 oC for 10 h. The amorphous TiO2 (TiO2-A) was obtained. Thermal annealing of the amorphous TiO2 at 250 oC, 350 oC, 550 oC and 750 oC for 4 h led to the production of TiO2 nanospheres, labeled as TiO2-250, TiO2-350, TiO2-550 and TiO2-750, respectively.

Reviewer 3 Report

The presented manuscript is devoted to the synthesis of antase/rutile with an antibacterial activity. The design of the investigation is quite simple, however, the application for the protection of paper is quite interesting.

I have some minor suggestions:

  1. The introduction should be revised to emphasize the novelty and originality of the study.
  2. Anatase to rutile ratio should be calculated using XRD data.
  3. Fig. 6. The error of measurements should be shown.

Author Response

Reviewer #3:

Response: Thank the reviewer very much for your instructions on the revision of our manuscript. We greatly appreciate the insightful comments of the reviewer of the paper. You have made some valuable suggestions that have led to great improvements to the manuscript.

1. The introduction should be revised to emphasize the novelty and originality of the study.

Response: Thanks for the reviewer’s suggestion. The introduction has been revised to emphasize the novelty and originality of the study.

2. Anatase to rutile ratio should be calculated using XRD data.

Response: Thanks for the reviewer’s suggestion. Anatase to rutile ratio has been calculated using XRD data. According to the peak areas of TiO2-750, the proportion of anatase (69.5%) and rutile (30.5%) phase is obtained.

3. Fig. 6. The error of measurements should be shown.

Response: Thanks for the reviewer’s good suggestion. The error bar of every group has been added (please see Fig. 6 in the revised manuscript). Error bars represent the standard deviation of three independent experiments.

Round 2

Reviewer 1 Report

This manuscript submitted to Nanomaterials focus on the characterization of TiO2 nanospheres with anatase/rutile heterophase junctions prepared via a template-free/annealing method. The material showed enhanced antibacterial activity against E. coli (98%) within 3 h simulated solar light irradiation. The work presents some novelty.  I thank the authors for their revised manuscript  and I found that it more profound and correct.
